# Microfluidic Deformability Study of an Innovative Blood Analogue Fluid Based on Giant Unilamellar Vesicles

**DOI:** 10.3390/jfb9040070

**Published:** 2018-12-04

**Authors:** Denise A. M. Carvalho, Ana Rita O. Rodrigues, Vera Faustino, Diana Pinho, Elisabete M. S. Castanheira, Rui Lima

**Affiliations:** 1Microelectromechanical Systems Research Unit (CMEMS-UMinho), DEI, University of Minho, Campus de Azurém, 4800-058 Guimarães, Portugal; denisemotacarvalho@gmail.com (D.A.M.C.); verafaustino@ipb.pt (V.F.); 2Centre of Physics, University of Minho, Campus de Gualtar, 4710-057 Braga, Portugal; ritarodrigues@fisica.uminho.pt; 3MEtRICs, Mechanical Engineering Department, University of Minho, Campus de Azurém, 4800-058 Guimarães, Portugal; 4CEFT, Faculty of Engineering of the University of Porto, Rua Dr. Roberto Frias, 4200-465 Porto, Portugal; 5Research Centre in Digitalization and Intelligent Robotics (CeDRI), Instituto Politécnico de Bragança, Campus de Santa Apolónia, 5300-253 Bragança, Portugal; diana@ipb.pt

**Keywords:** blood analogues, giant unilamellar vesicles, deformation index, in vitro blood, biomimetic, microcirculation

## Abstract

Blood analogues have long been a topic of interest in biofluid mechanics due to the safety and ethical issues involved in the collection and handling of blood samples. Although the current blood analogue fluids can adequately mimic the rheological properties of blood from a macroscopic point of view, at the microscopic level blood analogues need further development and improvement. In this work, an innovative blood analogue containing giant unilamellar vesicles (GUVs) was developed to mimic the flow behavior of red blood cells (RBCs). A natural lipid mixture, soybean lecithin, was used for the GUVs preparation, and three different lipid concentrations were tested (1 × 10^−3^ M, 2 × 10^−3^ M and 4 × 10^−3^ M). GUV solutions were prepared by thin film hydration with a buffer, followed by extrusion. It was found that GUVs present diameters between 5 and 7 µm which are close to the size of human RBCs. Experimental flow studies of three different GUV solutions were performed in a hyperbolic-shaped microchannel in order to measure the GUVs deformability when subjected to a homogeneous extensional flow. The result of the deformation index (DI) of the GUVs was about 0.5, which is in good agreement with the human RBC’s DI. Hence, the GUVs developed in this study are a promising way to mimic the mechanical properties of the RBCs and to further develop particulate blood analogues with flow properties closer to those of real blood.

## 1. Introduction

Blood is one of the most important human biofluids as it is an essential element for different body functions contributing to homeostasis. This complex fluid transports multiple substances important for the cellular tissue and organs of the human body [1]. 

For most blood studies, mainly rheological blood analogues are used because they have physical properties close to those of blood, such as viscosity, density, and refractive index [2]. The present study aims to contribute to the improvement in the knowledge of blood flow in arterioles and capillaries and the development of blood analogue fluids for in vitro blood experiments. Blood analogue fluids are extremely relevant, as working with real blood continues to have ethical, economical, and safety barriers. In fact, the development of blood analogue fluids continues to draw much attention from researchers around the world who are working to mimic the physical and rheological properties of real blood [1,2,3,4,5,6,7]. 

In the past, several blood flow studies were performed with Newtonian blood analogues composed of mixtures of glycerol and water [8,9,10,11]. Other studies have developed non-Newtonian fluids composed by xanthan gum or/and polyacrylamide diluted in glycerin and/or water [1,12]. However, in vitro blood experiments have found that it is crucial to take into account the cellular blood components that exist within the base fluid [13,14]. Pinto et al. [13], by performing experiments with in vitro blood samples, have observed a cell-free layer immediately downstream of a convergence bifurcation. Note that this physiological phenomenon does not happen with blood analogue fluids without solid elements, such as microparticles and microcapsules.

One of the major challenges in developing blood analogues is to incorporate cellular-like components able to perform fundamental functions, such as the transport of gases and nutrients, and the ability to deform under flow when they pass through a narrower capillary. Thus, one can not only think on the development of a fluid as a whole; it is also necessary to focus our attention on the cellular elements present in human blood and try to mimic the blood cells experimentally. Recent experimental works [5,15] developed blood analogue fluids containing polystyrene and polymethylmethacrylate particles in order to mimic the flow of Red Blood Cells (RBCs). However, these particles present a major drawback, which is their rigidity, and when they are flowing within microchannels, they do not deform in the same way as the RBCs. To overcome this limitation, the development of solid suspended elements able to mimic the rheological behavior of human RBCs is essential. In order to achieve this objective, an innovative blood analogue fluid containing giant unilamellar vesicles (GUVs) [16,17] was developed to mimic the flow of RBCs in microchannels. Giant unilamellar vesicles are simple membrane systems that are important in the study of more complex biological membranes; they contain heterogeneities in lipid composition, shape, chemical, and mechanical properties [18]. The GUV’s size (1–100 µm) and their geometry enable us to perform individual flow visualizations by means of an optical microscope, and as a result they have been applied in several biophysical contexts where membrane composition, tension, and geometry can be controlled and/or manipulated by using microscopic systems [18]. Here, the GUVs were prepared using a natural lipid mixture (soybean lecithin) by hydration of a lipid film, followed by extrusion through polycarbonate membranes, to guarantee a uniform vesicles diameter.

The main objective of this work is to develop an aqueous fluid containing GUVs with dimensions and mechanical properties similar to those of human RBCs. To investigate the ability of the GUVs to deform, flow measurements were performed in a microchannel with a hyperbolic constriction. Additionally, a rheological investigation of the proposed analogue fluid was performed by using a stress-controlled rheometer.

## 2. Materials and Methods

### 2.1. Preparation of GUVs

Giant unilamellar vesicle (GUV) aqueous dispersions were prepared using a natural lipid mixture, soybean lecithin (Sternchemie, Hamburg, Germany), through a lipid thin-film hydration method, as described [16]. All reagents used for the preparation of GUVs using the lipid film hydration are shown in Table 1. 

It has been reported that giant unilamellar vesicles are relatively stable over a period of weeks, at proper storage conditions (~4 °C) [19]. Tamba et al. [20], in a deep investigation focusing on the stability of this type of structure, showed that GUVs’ membranes in the liquid-ordered phase were stable and no leakage of encapsulated molecules occurred, even in the presence of a high concentration of Triton X-100, a well-known membrane disruption agent. 

First, a Tris-HCl buffer solution was prepared in a 250 mL volumetric flask, with 0.3025 g of Trizma^®^ Base (Sigma-Aldrich Inc., St. Louis, MO, USA) and ultrapure water (deionized water Millipore, Milli-Q grade, MilliporeSigma, St. Louis, MO, USA) to the final volume. A basic solution (pH around 10) was obtained, and then the pH was lowered to 7.4, using a hydrochloric acid solution. At the end, the buffer solution was stored in a glass flask.

In this method, three solutions of GUVs were prepared, with final lipid concentrations of C1 = 1 × 10^−3^ M, C2 = 2 × 10^−3^ M, and C3 = 4 × 10^−3^ M. For this purpose, the required amount (266 μL, 533 μL and 1066 μL) of a 5 × 10^−2^ M stock solution of soybean lecithin in chloroform was placed in a glass tube and dried under an ultrapure nitrogen stream, until a lipid film has formed. Then, 8 mL of the buffer solution previously prepared was added to the lipid film. The solutions thus obtained were placed in an ultrasonic bath (Branson 320) for 2 min, until a homogeneous dispersion was obtained. Then, three cycles of extrusion through polycarbonate membranes of 8-µm pore size were carried out (Extruder model: LIPEX™ 10 from Northern Lipids, Northern Lipids Inc., Burnaby, BC, Canada) using a pressure of 1.5 bar to improve size uniformity. A low pressure was applied to prevent bursting of the GUVs. Finally, the lipid dye Nile Red was added to the solutions to improve visualization of GUVs through microscopy. An ethanolic solution of Nile Red (2 × 10^−3^ M) was used. 

### 2.2. Microscopy System, Geometry, and Dimensions of the Microchannel

In this work we have used a high-speed microscopy system composed by a high-speed camera (Fastcam SA3, Photron, Tokyo, Japan), an inverted microscope (IX71, Olympus, Tokyo, Japan), a syringe pump (Harvard Apparatus PHD ULTRA, Holliston, MA, USA), and a syringe with a volume of 10 mL (Terumo, Tokyo, Japan). 

The GUVs’ flow measurements were performed in a microchannel with a hyperbolic constriction followed by a sudden expansion. This geometry is widely employed to perform separation of blood cells from plasma and to assess changes in cell deformability, due to its ability to achieve a controlled extensional flow at the center of the microchannel contraction and to measure large numbers of cells in one single run [5,21,22].

A soft-lithography technique was used to fabricate the microfluidic device. The length of the hyperbolic contraction region (C) was 600 µm, the width of the microchannel (L) was 400 µm, and the contraction width was 15 µm (see detail in Figure 1). Note that all microchannels have a depth of about 15 µm.

### 2.3. Image Analyses

After the images were obtained through an inverted microscopy system combined with a high-speed camera, image analysis was performed using the ImageJ^®^ software (Version 1.52i). Several measurements were made within the fabricated microchannels, such as GUVs’ volume fraction, diameter, and deformation.

### 2.4. Volume Fraction and Diameter of GUVs

To obtain the volume fraction of the produced GUVs for each concentration, images were first processed by using several functions from ImageJ^®^. To perform these measurements, a section of the channel was selected to carry out the study, as can be observed in Figure 1.

To determine the volume fraction of GUVs for each concentration, images were processed by using the ImageJ^®^ software. Briefly, the original video was first converted to a stack of images and then a background image was created containing static objects, such as microchannel walls and GUVs attached on the walls. This procedure was used to eliminate all the static objects and as a result only the flowing GUVs were measured. The images were then filtered and treated and the grey scale images were converted to binary images, adjusting the threshold level in order to obtain, at the end, images containing GUVs as black edged ellipsoidal objects against a white background (see Figure 1). After this segmentation process and by using the Wand and Measure functions, we were able to measure the area and diameter of all the flowing GUVs (see Figure 1). The following equation was used to calculate the GUVs’ volume fraction: (1)$$Volume\;fraction\;\left(\%\right)=\frac{Total\;Volume\;of\;GUVs}{Volume\;of\;selected\;section}\times 100$$
where the total volume of GUVs corresponds to the volume of all the GUVs located at a selected region, and volume of the selected section corresponds to W × L × depth of the microchannel (see Figure 1). Note that all GUVs were considered as perfect spheres. 

### 2.5. Deformation of GUVs

To perform deformability studies, the GUVs solutions C1, C2, and C3 were tested in a microfluidic device with a hyperbolic-shaped contraction. This kind of geometry has the ability to measure the degree of deformation of flowing objects under a controlled homogeneous extensional-flow field. By using this geometry, several studies have successfully measured red blood cells [21,22,23], white blood cells [24], and rigid particles [15]. Hence, in this study, we decided to use this geometry in order to assess the GUVs’ deformability under three different flow rates, i.e., 1, 5, and 10 μL/min. After the image acquisition by using a high-speed video microscopy system, the images were processed in a similar way to the procedure used to measure the area and diameter of the GUVs. Figure 2 shows two treated images, at two consecutive frames, for the solution C1 at a flow rate of 5 μL/min, flowing through the hyperbolic contraction. In Figure 2a, it possible to observe that the GUV elongates when it is inside the contraction due to the high extensional flow, whereas in Figure 2b the GUV returns close to its original shape at rest (circular shape) due to the low shear flow that the GUV is submitted to at this region of the microchannel.

To investigate the deformation of GUVs flowing along the microchannel, the major (*Major*) and minor (*Minor*) dimensions of the GUVs were measured in order to calculate their deformation index (*DI*) by using the following equation: (2)$$DI=\frac{\left(Major-Minor\right)}{\left(Major+Minor\right)}$$
where *Major* and *Minor* correspond to the major and minor axis lengths of the ellipse that can be best fitted to the GUVs. Note that the DI values are between 0 and 1, i.e., 0 corresponds to a perfect circle and higher values mean more deformed shapes.

### 2.6. Rheological Characterization

The rheological characterization of the GUV solutions was performed by a stress control rheometer (Bohlin CVO, Malvern, Worcestershire, UK) using a cone-plate geometry with a diameter of 55 mm and an angle of 1. The gap size for this geometry was 0.03 mm. The steady shear viscosity curves for all GUV solutions were obtained in a range of shear rates above 10 until 10,000 s^−1^. The same test was performed with a blood sample of 5% hematocrit (Hct) in a saline solution [5,15,25]. All tests were carried out at a temperature of 22 ± 1 °C, corresponding to the room temperature verified during the performance of the flow studies. In addition, the minimum torque line [26], that represents the lower limit of the equipment, was added to the data in order to be considered for accurate measurements. 

## 3. Results and Discussion

### 3.1. Volume Fraction of GUVs

The volume fraction of the GUVs obtained for each of the solutions with different concentrations is shown in Table 2. By analyzing this table, it can be concluded that, as expected, there is an increase in the volume fraction of GUVs with increasing concentration of the lipid mixture. However, the increase in volume fraction is not proportional to the rise in concentration, indicating that other factors, like the variation in GUV size, are likely to influence the volume fraction of GUVs. In fact, each GUV contains a large number of lipid molecules, and a relatively small variation in GUV diameter may have significant influence in the number of lipid molecules per vesicle. Moreover, high lipid concentrations may promote the formation of bilayer fragments in solution, with a decrease in the volume fraction of GUVs present in the solution.

### 3.2. Diameter of GUVs

Figure 3 shows the diameters of the GUVs obtained for each of the solution concentrations. As can be seen in Figure 3, for all concentrations, there are a high number of GUVs with diameters between 5 and 7 µm. These results show that the proposed protocol allows for the production of GUVs with sizes close to those of the RBCs, since human RBCs have diameters between 6 and 9 µm [27]. Additionally, these results indicate that the higher the lipid concentration in solution is, the lower are the average diameters of the GUVs and the higher is the dispersity in size values. Thus, the solution that contains GUVs closest in terms of size with human RBCs is the one with the lower lecithin concentration, i.e., solution C1 (1 × 10^−3^ M).

### 3.3. Deformation of GUVs

The study of the GUVs’ deformability was performed in seven different regions along the microchannel, as can be seen in Figure 4. The deformation of the GUVs was investigated for the developed solutions (C1, C2, and C3) at three different flow rates, i.e., 1, 5, and 10 μL/min. Note that at each flow rate we measured about 500 GUVs, i.e., for each region we measured the DI of at least 70 GUVs. 

From Figure 4, it can be observed that the GUVs have the ability to deform in the region of the hyperbolic contraction and, as expected, the DI increases as the flow rate rises. The GUVs upstream of the contraction have a DI close to zero, mostly due to the low shear rate. However, as they approach the hyperbolic contraction, the GUVs’ DI increases significantly. After the contraction, this index decreases until it returns to values close to zero. These results are extremely promising, since GUVs, similarly to RBCs, have the ability to deform under strong extensional flows. 

Recently, Pinho and colleagues [15] measured the DI of human RBCs with geometry and flow conditions similar to the ones used in this study. They measured the DI of human RBCs at a hyperbolic contraction region of about 0.36 for a flow rate of 5 μL/min (see Figure 5). For the same flow rate, the values of the GUVs’ DI obtained in this study were about 0.48, which are in reasonable agreement with the human RBCs’ DI. Additionally, in Figure 5 it is possible to observe that the results were in closer agreement at the downstream region of the contraction where the cells and GUVs tend to recover to their original shape. Hence, it is evident that the GUVs produced in this study are a promising way to mimic the mechanical properties of the RBCs and, consequently, to develop blood analogues with flow properties closer to real blood.

### 3.4. Rheology of GUVs

To perform the rheology study, the three proposed solutions, C1, C2, and C3, were analyzed by using a stress-controlled rheometer. In this study, a small amount of each solution was placed below the cone-plate geometry and all the shear flow curves were obtained at a temperature of about 22 °C. Figure 6 shows the rheological results for the three developed solutions. Analyzing Figure 6, it can be observed that the three solutions have a similar viscosity. However, a slight increase with the increment of the concentration of soybean lecithin was observed.

More evidence from Figure 6 is that the viscosity of the three solutions of GUVs is close to the viscosity of water, which is 0.0010002 Pa·s at a temperature of 20 °C. This result was expected, since the medium of GUV solutions is an aqueous buffer solution (main component is ultrapure water). Next, we decided to perform a rheological comparison between the solution C1 and a sample with 5% hematocrit in saline solution (see Figure 7).

The results from Figure 7 show that the solution C1 has a viscosity close to the 5% haematocrit in a saline solution. Thus, this comparison shows that the blood analogue proposed in this work is in close agreement with rheological behavior of in vitro blood samples having low Hct. These kind of samples are widely used to determine the rate of deformability of different cell types, such as healthy cells, cells infected with the malaria parasite, and cancer cells [21,28,29].

## 4. Limitations and Future Directions

In this study, low concentrations of GUVs were used in order to better follow individual GUVs and measure their deformability along the hyperbolic microchannel. By using high concentrations, it is extremely difficult to follow the behavior of individual GUVs, as most of the time the GUVs are surrounded by neighboring GUVs. For larger Htcs, it is well known that in microcirculation there is a formation of a cell-free layer around the microvessel walls. In this way, we also performed measurements with samples having high concentrations of GUVs. However, in contrast to the RBCs, the GUVs were extremely difficult to visualize and as result the cell free layer measurements need to be improved in the near future. The visualization of GUVs can be improved by two different approaches:
-Fluorescent water-soluble dyes (e.g., Rhodamine B, fluorescein) can be encapsulated into GUVs to color their aqueous inner compartment (a size exclusion chromatography will allow the separation of the non-encapsulated dye molecules);-Phospholipids labelled with fluorescent molecules (e.g., with Rhodamine B, dansyl, or cyanine dyes) can be included in GUVs’ formulations to more effectively color the lipid bilayer.

The GUVs deformability, when compared with human RBCs, indicates that GUVs need to be slightly less deformable. Hence, to increase GUVs rigidity, the use of lipids with a high melting transition temperature, *T*_m_, in GUVs’ composition could be a solution. These lipids are in the rigid gel phase at room temperature, causing a decrease in fluidity of the lipid bilayer. Considering that phosphatidylcholines are the most abundant lipids in biological membranes, the lipids DPPC (dipalmitoylphosphatidylcholine), with *T*_m_ = 41 °C [30], or DSPC (distearoylphosphatidylcholine, *T*_m_ = 55 °C [30]) might be a good choice. Another possibility is the inclusion of cholesterol, which promotes the formation of an ordered liquid phase in the lipid bilayer when it is in the fluid disordered state [31], enhancing the mechanical stability of the membrane [32]. A different strategy, consisting of the inclusion of magnetic nanoparticles in the GUVs’ formulation (forming magnetic GUVs), could also be attempted. It has been reported that magnetic nanoparticles tend to aggregate in membranes of liposomes and cells [33], and this effect could be explored to increase the rigidity of the GUVs’ lipid bilayer. 

## 5. Conclusions

The main objective of this work was to develop a particulate blood analogue fluid based on GUVs, with their mechanical properties comparable to RBCs flowing in microchannels.

In this work, the volume fraction was determined for three solutions of GUVs (from natural soybean lecithin), differing from one another by the lipid concentration. It has been observed that the higher the amount of lecithin, the higher the GUVs’ volume fraction in the fluid. Regarding GUVs diameters, it was found that in all solutions, the GUVs size varied between 5 and 7 µm, which is close to the size of human RBCs. In addition, it was found that the higher the concentration of soybean lecithin, the lower the average diameter of GUVs, together with the larger dispersity in size values.

Experimental measurements of the produced GUVs flowing through hyperbolic-shaped microchannels have shown not only that the GUVs had the highest deformation at the end of the contraction region, due to the strong extensional flow, but also the GUVs’ DI increased with the increasing flow rate. Additionally, when the GUVs were subjected to a strong extensional flow, it was found that GUVs’ DIs were about 0.36, which are in reasonable good agreement with the human RBCs’ DI. These results make them a promising candidate to mimic the RBCs’ mechanical properties and, consequently, to develop particulate solutions with flow properties analogous to real whole blood.

Regarding rheological behavior, the GUV solutions have shown similar viscosities for all the tested concentrations. Nevertheless, a slight increase with lipid concentration was observed. The solution 1 × 10^−3^ M in lecithin revealed a viscosity similar to the one of a 5% Hct in saline solution. 

Generally, the results obtained in this work clearly indicate that the introduction of giant unilamellar vesicles into non-particulate blood analogue fluids may enable a better mimicking of real blood. In future studies will likely involve testing some modifications on lipid composition, surface charge, and preparation methods, to advance towards a blood analogue fluid that fully mimics all blood cell components and flow behavior.

## Figures and Tables

**Figure 1 jfb-09-00070-f001:**
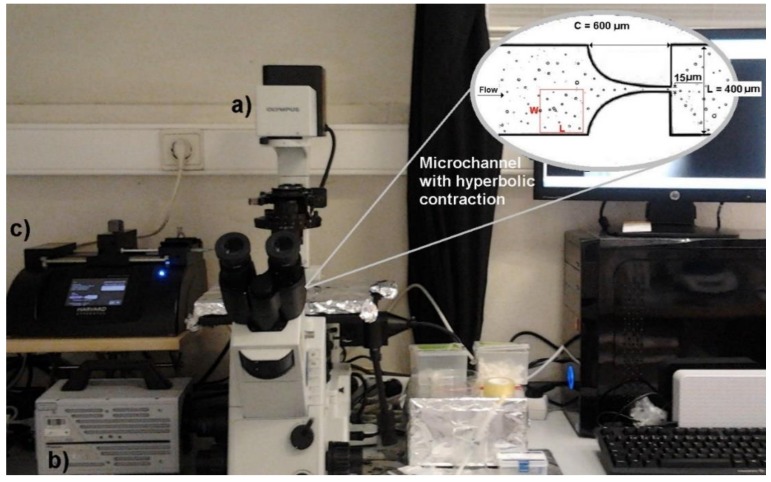
Equipment used for the measurement and visualization of the flows: (**a**) inverted microscope; (**b**) high-speed camera; (**c**) syringe pump. In addition, in the upper right side there is an image obtained through ImageJ software of the volume fraction of GUVs, where W = 200 µm and L= 200 µm. The main dimensions of the hyperbolic microchannel are shown at the detailed image.

**Figure 2 jfb-09-00070-f002:**
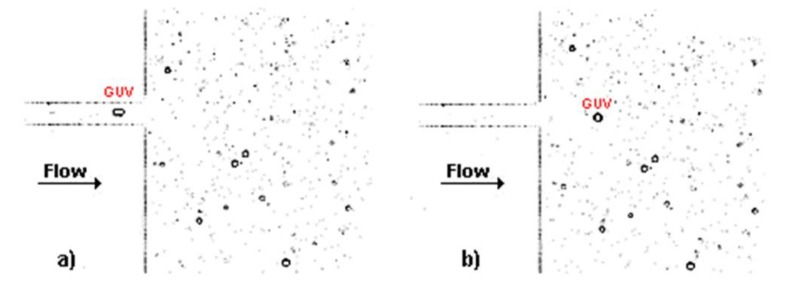
Images obtained through the ImageJ software to study the deformation of GUVs: (**a**) GUV within the hyperbolic contraction; (**b**) GUV at the expansion region. These images correspond to the solution C1 at a flow rate of 5 μL/min.

**Figure 3 jfb-09-00070-f003:**
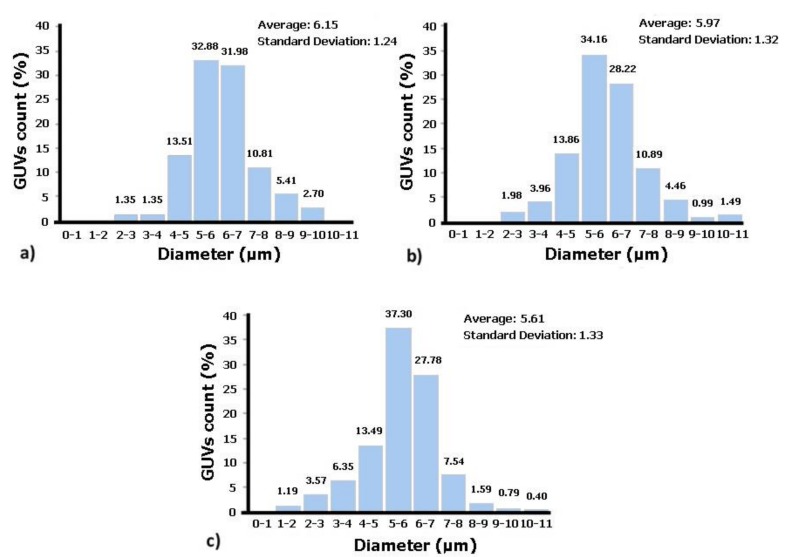
Diameter of GUVs. (**a**) Solution C1; (**b**) Solution C2; (**c**) Solution C3.

**Figure 4 jfb-09-00070-f004:**
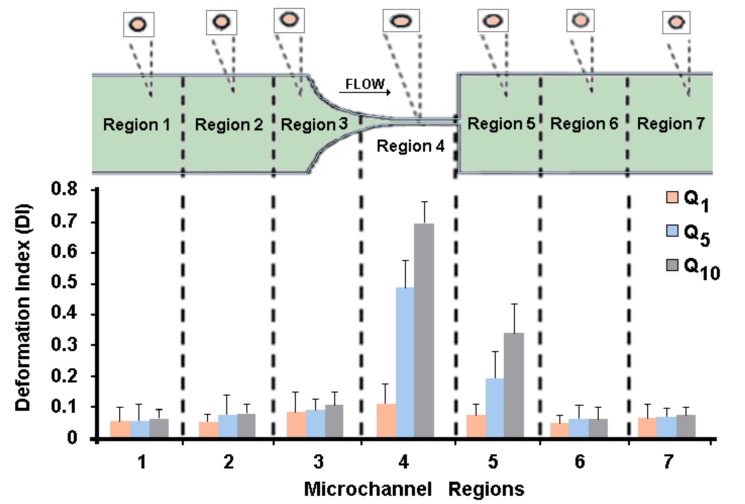
Deformation index (DI) of GUVs along the microchannel for the solution C1 (1 × 10^−3^ M) and for the flow rates Q1 = 1 μL/min, Q5 = 5 μL/min, and Q10 = 10 μL/min.

**Figure 5 jfb-09-00070-f005:**
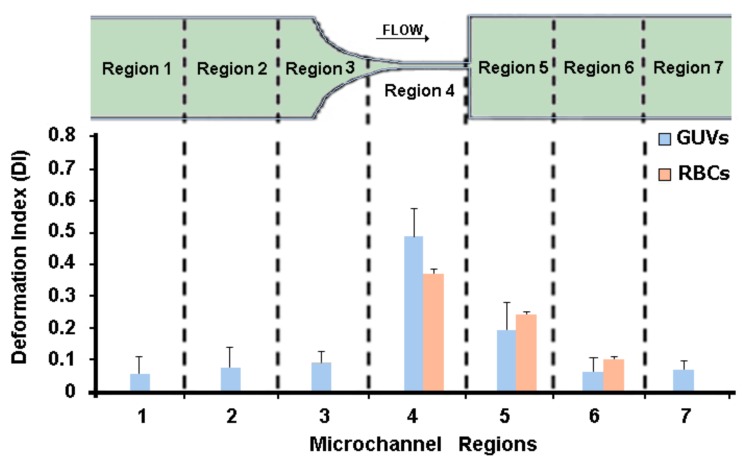
Deformation index (DI) of GUVs (C1) and human red blood cells (RBCs) at different microchannel regions with a flow rate of 5 μL/min.

**Figure 6 jfb-09-00070-f006:**
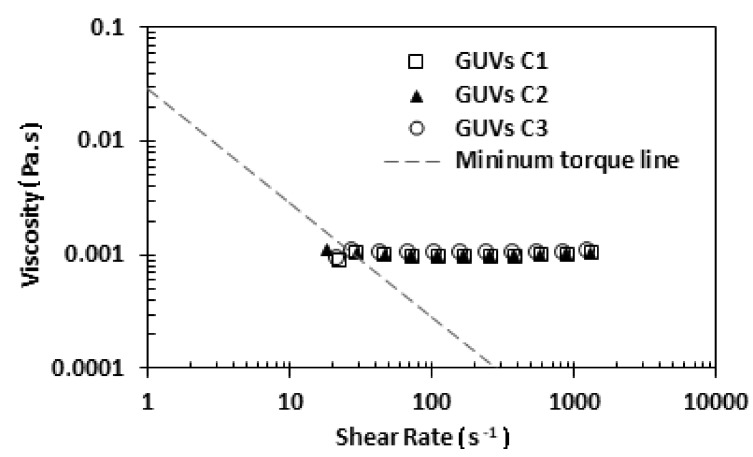
Viscosity of the proposed GUV solutions: viscosity as a function of shear rate. The dashed line represents the minimum torque line of the equipment.

**Figure 7 jfb-09-00070-f007:**
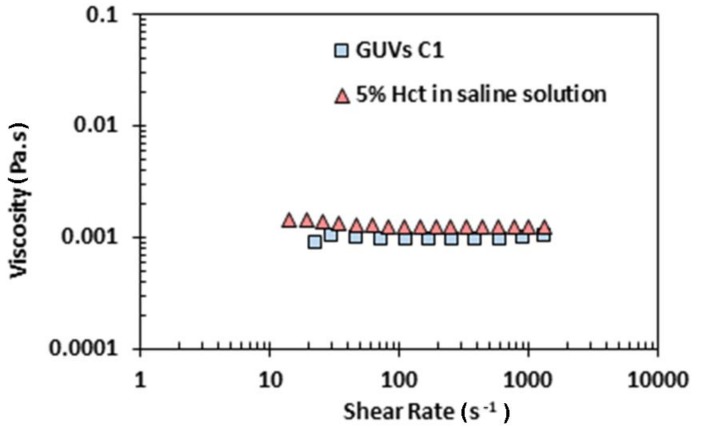
Viscosity of the solution of GUVs C1 and 5% Hct in saline solution.

**Table 1 jfb-09-00070-t001:** Reagents for the giant unilamellar vesicle (GUV) preparation by using the lipid film hydration method.

Compound	Chemical Formula	Molecular Weight (g mol^−1^)	Purity	Manufacturer
Soybean lecithin	C_42_H_84_NPO_9_	776.00	N.A. *	Sternchemie (Hamburg, Germany)
Chloroform	CHCl_3_	119.38	99.8%	Sigma-Aldrich^®^ (St. Louis, MO, USA)
Trizma^®^ base	C_4_H_11_NO_3_	121.14	99.9%	Sigma-Aldrich^®^ (St. Louis, MO, USA)
Nile Red	C_20_H_18_N_2_O_2_	318.37	N.A.*	Sigma-Aldrich^®^ (St. Louis, MO, USA)

* N.A.: Not available.

**Table 2 jfb-09-00070-t002:** Volume fraction of GUVs at different concentrations.

Concentration (M)	Volume Fraction (%)
1 × 10^−3^	1.6
2 × 10^−3^	2.1
4 × 10^−3^	2.5

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
