# Peer review of "Microfluidic Deformability Study of an Innovative Blood Analogue Fluid Based on Giant Unilamellar Vesicles"

_jfb, 2018, doi:10.3390/jfb9040070_

Round 1
Reviewer 1 Report
Dear Editor
In the article entitled "Microfluidic deformability study of an innovative blood analog fluid based on giant unilamellar vesicles," authored by Denise A. M. Carvalho et al., the authors discussed the usefulness of Giant Unilamellar Vesicles (GUVs) for mimicking the rheological properties of RBC.
The subject of the presented article is the development of an aqueous fluid containing GUVs with dimensions and mechanical properties similar to human RBCs. The authors propose a new construction for solving this problem.
The authors prepared a very informative manuscript on this topic. The text is sound, and the manuscript is well written.
However, there are several essential points that authors should consider before publishing.
Major:
1. The central subject of the presented article is the development of an aqueous fluid containing GUVs with deformability properties similar to human RBCs. However, the authors in the present article do not carry out the results of parallel measurements of erythrocytes and GUVs deformation on the described microscopy system.
2. Stability of GUVs must be discussed.
3. Authors must submit a section titled "Study Limitation."
Figures:
1. Fig. 4 must be removed, and the data must be presented in the text.
2. There is no need to show graphs for all three lipid concentrations (Figs. 6-8), please select one of them.
3. Fig. 9b is a direct consequence of Fig. 9a, so there is no point in publishing the Fig. 9b.
Text:
Line 110: Model of syringe pump must be described.
Lines 185-187: The explanation given here is poorly justified.
Author Response
In the article entitled "Microfluidic deformability study of an innovative blood analog fluid based on giant unilamellar vesicles", authored by Denise A. M. Carvalho et al., the authors discussed the usefulness of Giant Unilamellar Vesicles (GUVs) for mimicking the rheological properties of RBC.
The subject of the presented article is the development of an aqueous fluid containing GUVs with dimensions and mechanical properties similar to human RBCs. The authors propose a new construction for solving this problem.
The authors prepared a very informative manuscript on this topic. The text is sound, and the manuscript is well written.
However, there are several essential points that authors should consider before publishing.
Major:
1. The central subject of the presented article is the development of an aqueous fluid containing GUVs with deformability properties similar to human RBCs. However, the authors in the present article do not carry out the results of parallel measurements of erythrocytes and GUVs deformation on the described microscopy system.
Following reviewer’s suggestion, we have included a new figure (Figure 5) showing the deformability of both GUVs and human RBCs under similar flow conditions. Note that the results performed with RBCs were obtained from other research project under progress where we were only interested to study regions 4, 5 and 6. From the figure below it is possible to observe that there is reasonable good agreement. However, we recognize that we still need to improve the mechanical properties of GUVs especially at the region 4 where the GUVs have more elongation when compared with human RBCs. These new results and future work were added in the revised manuscript at the section 3.3 Deformation of GUVs and 4. Limitations and future directions.
2. Stability of GUVs must be discussed.
We acknowledge reviewer’s suggestion and the following explanation was introduced in the revised manuscript (lines 88-92): “It has been reported that Giant Unilamellar Vesicles are relatively stable over a period of weeks, at proper storage conditions (~ 4°C) [19]. Tamba et al. [20], in a deep investigation focusing the stability of this type of structures, showed that GUVs membranes in the liquid-ordered phase were stable and no leakage of encapsulated molecules occurred, even in the presence of a high concentration of Triton X-100, a well-known membrane disruption agent.”
New references [19] and [20] were added to the revised manuscript.
3. Authors must submit a section titled "Study Limitation."
Following reviewer’s suggestion, a section titled “Limitations and future directions” was added to the revised version (lines 280-307).
“In this study low concentrations of GUVs were used in order to better follow individual GUVs and measure their deformability along the hyperbolic microchannel. By using high concentrations it is extremely difficult to follow the behavior of individual GUVs as most of the times the GUVs are surrounded by neighborhood GUVs. For larger Htcs it is well known that in microcirculation there is a formation of a cell free layer around the microvessel walls. In this way, we have also performed measurements with samples having high concentrations of GUVs. However, in contrast to the RBCs, the GUVs were extremely difficult to visualize and as result the cell free layer measurements need to be improved in the near future. The visualization of GUVs can be improved by two different approaches:
- Fluorescent water-soluble dyes (e.g. Rhodamine B, fluorescein) can be encapsulated in GUVs to color their aqueous inner compartment (a size exclusion chromatography will allow the separation of the non-encapsulated dye molecules);
- Phospholipids labelled with fluorescent molecules (e.g. with Rhodamine B, dansyl or cyanine dyes) can be included in GUVs formulations to more effectively color the lipid bilayer.
The GUVs deformability, when compared with human RBCs, indicates that GUVs need to be slightly less deformable. Hence, to increase GUVs rigidity, the use of lipids with a high melting transition temperature, Tm, in GUVs composition could be a solution. These lipids are in the rigid gel phase at room temperature, causing a decrease in fluidity of the lipid bilayer. Considering that phosphatidylcholines are the most abundant lipids in biological membranes, the lipids DPPC (dipalmitoylphosphatidylcholine), with Tm = 41 °C [30], or DSPC (distearoylphosphatidylcholine, Tm = 55 °C [30]) can be a good choice. Another possibility is the inclusion of cholesterol, which promotes the formation of an ordered liquid phase in the lipid bilayer when it is in the fluid disordered state [31], enhancing the mechanical stability of the membrane [32]. A different strategy, consisting in the inclusion of magnetic nanoparticles in the GUVs formulation (forming magnetic GUVs), could also be attempted. It has been reported that magnetic nanoparticles tend to aggregate in membranes of liposomes and cells [33], and this effect could be explored to increase the rigidity of GUVs lipid bilayer.”
Figures:
1. Fig. 4 must be removed, and the data must be presented in the text.
Figure 4 was removed and the data included in Table 2.
2. There is no need to show graphs for all three lipid concentrations (Figs. 6-8), please select one of them.
Figures 6-8 were transformed in only one figure (new Figure 4).
3. Fig. 9b is a direct consequence of Fig. 9a, so there is no point in publishing the Fig. 9b.
The figure (Figure 6 in the revised version) was modified according to the suggestion.
Text:
Line 110: Model of syringe pump must be described.
The model of the syringe pump was added in the revised manuscript (lines 113-114).
Lines 185-187: The explanation given here is poorly justified.
A more complete explanation was added in the revised manuscript (now in lines 200-205): “However, the increase in volume fraction is not proportional to the rise in concentration, indicating that other factors, like the variation in GUVs size, are likely to influence the volume fraction of GUVs. In fact, each GUV contains a large number of lipid molecules and a relatively small variation in GUVs diameter may have significant influence in the number of lipid molecules per vesicle. Moreover, high lipid concentrations may promote the formation of bilayer fragments in solution, with a decrease in the volume fraction of GUVs present in the solution.”
Reviewer 2 Report
In this manuscript, the authors use Giant Unilamellar Vesicles (GUVs) to mimic RBCs and their solutions as blood analogues. A microdevice with a constriction channel, which has been reported previously (as cited in Ref. 1), is employed to study the deformability of GUVs; and the rheology has also been explored. As the results demonstrated, the similarity of this blood analogue is surprisingly in good agreement with real blood samples in both deformability and viscosity. It is recommended for publication with additional revisions.
Overall, the figure resolution needs to be improved.
Fig.1 is not necessary except the inset showing the dimension of the device, which can be combined with Fig.2.
It is difficult to compare the bar heights in DIs of GUVs in three separated figures (Fig. 6,7,8). It is suggested to use color codes to plot them in one. How many total GUVs were measured in this set of experiment?
Could this device and method be used to mimic blood samples with higher hematocrit?
Author Response
In this manuscript, the authors use Giant Unilamellar Vesicles (GUVs) to mimic RBCs and their solutions as blood analogues. A microdevice with a constriction channel, which has been reported previously (as cited in Ref. 1), is employed to study the deformability of GUVs; and the rheology has also been explored. As the results demonstrated, the similarity of this blood analogue is surprisingly in good agreement with real blood samples in both deformability and viscosity. It is recommended for publication with additional revisions.
Overall, the figure resolution needs to be improved.
The resolution of the figures was improved as much as possible.
1. Fig. 1 is not necessary except the inset showing the dimension of the device, which can be combined with Fig. 2.
As suggested by the reviewer, previous figures 1 and 2 were combined in the new figure 1.
2. It is difficult to compare the bar heights in DIs of GUVs in three separated figures (Fig. 6,7,8). It is suggested to use color codes to plot them in one. How many total GUVs were measured in this set of experiment?
The figure (new figure 4) was modified and improved according to the reviewer’s suggestion. For each flow rate it was measure about 500 GUVs that means for each region we have measured the DI of at least 70 GUVs. This information was added in the revised manuscript (lines 224-226).
3. Could this device and method be used to mimic blood samples with higher hematocrit?
The following explanation was included in the new section “Limitations and future directions”: “In this study low concentrations of GUVs were used in order to better follow individual GUVs and measure their deformability along the hyperbolic microchannel. By using high concentrations it is extremely difficult to follow the behavior of individual GUVs as most of the times the GUVs are surrounded by neighborhood GUVs. For larger Htcs it is well known that in microcirculation there is a formation of a cell free layer around the microvessel walls. In this way, we have also performed measurements with samples having high concentrations of GUVs. However, in contrast to the RBCs, the GUVs were extremely difficult to visualize and as result the cell free layer measurements need to be improved in the near future. The visualization of GUVs can be improved by two different approaches:
- Fluorescent water-soluble dyes (e.g. Rhodamine B, fluorescein) can be encapsulated in GUVs to color their aqueous inner compartment (a size exclusion chromatography will allow the separation of the non-encapsulated dye molecules);
- Phospholipids labelled with fluorescent molecules (e.g. with Rhodamine B, dansyl or cyanine dyes) can be included in GUVs formulations to more effectively color the lipid bilayer.
The GUVs deformability, when compared with human RBCs, indicates that GUVs need to be slightly less deformable. Hence, to increase GUVs rigidity, the use of lipids with a high melting transition temperature, Tm, in GUVs composition could be a solution. These lipids are in the rigid gel phase at room temperature, causing a decrease in fluidity of the lipid bilayer. Considering that phosphatidylcholines are the most abundant lipids in biological membranes, the lipids DPPC (dipalmitoylphosphatidylcholine), with Tm = 41 °C [30], or DSPC (distearoylphosphatidylcholine, Tm = 55 °C [30]) can be a good choice. Another possibility is the inclusion of cholesterol, which promotes the formation of an ordered liquid phase in the lipid bilayer when it is in the fluid disordered state [31], enhancing the mechanical stability of the membrane [32]. A different strategy, consisting in the inclusion of magnetic nanoparticles in the GUVs formulation (forming magnetic GUVs), could also be attempted. It has been reported that magnetic nanoparticles tend to aggregate in membranes of liposomes and cells [33], and this effect could be explored to increase the rigidity of GUVs lipid bilayer.”
Round 2
Reviewer 1 Report
The authors made the necessary changes to the text of the article, in this revised form it can be published.